fluid mechanics

drops, hydrostatics, surface tension, Young–Laplace

**Author for correspondence:**
Amir H. Fatollahi
e-mail: fath@alzahra.ac.ir

# Identities for droplets with circular footprint on tilted surfaces

François Dunlop[1], Amir H. Fatollahi[2], Maryam Hajirahimi[3] and Thierry Huillet[1]

[1]Laboratoire de Physique Théorique et Modélisation, CY Cergy Paris Université, CNRS UMR 8089, 95302 Cergy-Pontoise, France
[2]Department of Physics, Faculty of Physics and Chemistry, Alzahra University, Tehran 1993891167, Iran
[3]Physics Group, South Tehran Branch, Islamic Azad University, P. O. Box 11365, Tehran 4435, Iran

AHF, 0000-0001-6330-4085

Exact mathematical identities are presented between the relevant parameters of droplets displaying circular contact boundary based on flat tilted surfaces. Two of the identities are derived from the force balance, and one from the torque balance. The tilt surfaces cover the full range of inclinations for sessile or pendant drops, including the intermediate case of droplets on a wall (vertical surface). The identities are put under test both by the available solutions of a linear response approximation at small Bond numbers as well as the ones obtained from numerical solutions, making use of the *Surface Evolver* software. The subtleties to obtain certain angle-averages appearing in identities by the numerical solutions are discussed in detail. It is argued how the identities are useful in two respects. First is to replace some unknown values in the Young–Laplace equation by their expressions obtained from the identities. Second is to use the identities to estimate the error for approximate analytical or numerical solutions without any reference to an exact solution.

## 1. Introduction

Understanding the skewed shape of a sessile drop pinned on a flat incline has a long history in Physics, starting with [1,2]. It rises new asymmetrical problems compared to more studied situations where the substrate is horizontal. In principle, such problems can be handled while making use of the Young–Laplace nonlinear partial differential equation, translating a balance between surface tension forces and gravity acting on the drop. See [3] and references therein, where a perturbative approach to this problem at small Bond number was addressed when the footprint of the droplet is

held fixed and circular. More recently, the case of pendant drops has also attracted some interest, see [4–7] and references therein. A perturbative approach to this problem at small Bond number has also been addressed in [8], in a similar set-up. An empirical relation between incline slope angle and contact angles at the front and rear of the droplet was given by [9], and further studied by many authors, see [3,10,11] and references therein. It relies on an approximation of the balance of forces equation along the substrate, at small Bond number. Balance of forces normal to the substrate also deserves interest, together with relations arising from the torque balance. The three-phase contact angles (for the various azimuthal angles) of a liquid condensed on a substrate are in direct relation with interfacial and body forces acting on sessile or pendant drops.

Despite the settled role of the droplets based on different surfaces, the cases for which there are exact solutions are rare. During more than a century, different numerical and analytical methods have been developed for more efficient and finer approach to drop's profile for cases with no exact solution. To evaluate or rate these numerical and analytical methods some criteria are needed, among which are the exact mathematical relations between the relevant parameters of the problem. An early example of these identities is the one by [12] between the volume, curvature at apex, height and contact radius of axi-symmetric drops on a flat horizontal surface (see also [13,14]). For axi-symmetric drops on curved surfaces the very same identities are derived in [15]. For droplets under the combined tangential and normal body forces the dynamical relations between shape parameters have been presented in a linear approximation recently [16].

This issue has also been studied semi-analytically in [17], where the problem of understanding the contact line evolution of slender unpinned droplets under arbitrary scenarios of forces is addressed, based on experimentally observed contact lines. Related to this point, in [18], sessile droplets at different tilting angles are experimentally subject to varying centrifugal forces in order to explore their spreading/sliding behaviour for different volumes and initial shapes (including non-axisymmetric). In particular, a test of the applicability of the Furmidge equation for the retention force is discussed.

The mathematical identities are important in another respect, that is reducing the initial unknown values of the problem. This in particular proves helpful because some of these unknown values appear in the first place in the differential equation governing the profile of the drop. Whether one tries a perturbative solution of the drop's profile or a numerical one, reducing the initially unknown values facilitates or boosts the procedure of reaching the final result. As an example, in [14] the identity is used to replace a combination of the unknown apex curvature and height in the Young–Laplace equation, in the procedure of developing a perturbative solution for lightweight drops. As another example, in a numerical solution of axi-symmetric drops with fixed contact angle one may use contact radius or apex height as a starting point and the other one as the final point. But the trouble is that both values are unknown in the first place, and in principle one has to search time-consumingly a two-dimensional parameter space to find both values that match the solution. However, by the identity and thanks to the mentioned combination, one can replace the height by the radius, reducing the procedure to a simple one-parameter shooting method [19].

The main purpose of the present work is to highlight examples of mathematical identities for droplets with circular contact boundary based on flat tilted surfaces. The identities are derived based on the force balance along parallel and normal directions of the tilted surface, as well as the torque balance of the droplet. The force balance normal to the tilted surface generalizes that of [12–14] for a horizontal substrate. The identity along the tilted surface is in fact an exact version of the approximate empirical Furmidge relation [9]. The identity by the torque balance is apparently the one that is introduced here, and remains to be verified numerically. All three identities are checked at the first-order approximation of the Bond number, the so-called linear response ansatz [3]. For the identities by the force balance various numerical tests are provided, generated by the *Surface Evolver* as a vertex-edge-facet element software [20].

The organization of the rest of the work is as follows. In §2 the two identities by the force balance along and normal to the surface are derived. In §3 the identity by the torque-balance condition on the droplet is derived. The check of all identities at linear response approximation is presented in §4. Numerical checks of force-balance identities are presented in §5. This requires the contact angle as a function of azimuth angle and certain averages over it, which is the subject of §6. In §7 the possible use of identities is illustrated. Section 8 is devoted to concluding remarks.

# 2. Identities by force balance

The set-up for the pinned droplet on a tilted plane is as follows. The $z$-axis is perpendicular to the substrate and inward to the liquid, with $x$-axis along the slope downward. For the droplet with

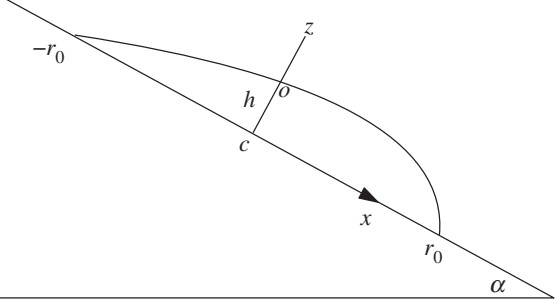

**Figure 1.** Set-up of a droplet on a flat tilted surface.

contact-circle of radius $r_0$, the origin is set to be the centre of circle, $c$. The angle of the substrate with the horizontal direction is $\alpha \in [0, \pi]$, with $\alpha = \pi$ representing the case of a drop pendant from the ceiling. The set-up is summarized in figure 1. We use the polar angle $\varphi$ on the substrate, with $\varphi = 0$ representing the x-axis, as usual. The hydrostatic pressure inside the contact-circle depends only on $x$, given by

$$p(x) = p_c + \rho g x \sin \alpha \tag{2.1}$$

with $\rho$ the density of the drop, and $p_c$ the pressure at the centre of the contact-circle.

## 2.1. Force balance along surface

First we consider the more familiar identity, stemming from force balance along the substrate. The capillary force along the $x$ direction reads

$$F_x = \gamma \int_{-\pi}^{\pi} r_0 \, \mathrm{d}\varphi \, \cos \theta_\alpha(\varphi) \, \cos \varphi \tag{2.2}$$

with $\theta_\alpha(\varphi)$ the contact-angle of the liquid-substrate at polar angle $\varphi$ on a slope with angle $\alpha$. By symmetry $\theta_\alpha(\varphi) = \theta_\alpha(-\varphi)$, so we expect the following Fourier expansion:

$$\cos \theta_\alpha(\varphi) = C_0 + \sum_{n=1}^{\infty} C_n \cos(n\varphi), \tag{2.3}$$

where

$$C_0 = \frac{1}{2\pi} \int_{-\pi}^{\pi} \mathrm{d}\varphi \cos \theta_\alpha(\varphi) = \langle \cos \theta_\alpha(\varphi) \rangle \tag{2.4}$$

and

$$C_n = \frac{1}{\pi} \int_{-\pi}^{\pi} \mathrm{d}\varphi \cos \theta_\alpha(\varphi) \cos(n\varphi), \quad n \geq 1 \tag{2.5}$$

leading to

$$F_x = \gamma \pi r_0 \, C_1. \tag{2.6}$$

The force balance along the surface then gives:

$$mg \sin \alpha + \gamma \pi r_0 \, C_1 = 0 \tag{2.7}$$

with $m = \rho V$ as the mass of the droplet with volume $V$. The dimensionless form of (2.7) is

$$2 \, \mathrm{Bo} \sin \alpha + \pi C_1 = 0, \tag{2.8}$$

where Bo is the modified Bond (or Eötvös) number taken as $\mathrm{Bo} = mg/(2r_0\gamma)$. Consistently with [21], our use of Bo contains only the input parameters of the drop-solid system, namely volume of the drop, gravitational acceleration, fluid density, surface tension, and radius of the footprint and not data involving the dimensions of the drop unknown prior to the experiment (such as the height of the drop or radius of curvature at the drop apex).

The coefficient $C_1$, equal to twice the average over $\varphi$ of $\cos\theta_\alpha(\varphi)\cos\varphi$, is remarkably linear in Bo as will be seen later.

In §5, in order to confront theoretical and simulation values, we will evaluate and compare to 1 the ratio

$$\text{ratio}_\parallel = -2\,\text{Bo}\,\sin\alpha/(\pi C_1). \tag{2.9}$$

## 2.2. Force balance normal to surface

The capillary force normal to the surface (downward $z$ direction) reads

$$F_z = \gamma \int_{-\pi}^{\pi} r_0\,\mathrm{d}\varphi\,\sin\theta_\alpha(\varphi). \tag{2.10}$$

Again by symmetry we expect $\theta_\alpha(\varphi) = \theta_\alpha(-\varphi)$, hence the Fourier expansion

$$\sin\theta_\alpha(\varphi) = A_0 + \sum_{n=1}^{\infty} A_n \cos(n\varphi), \tag{2.11}$$

$$A_0 = \frac{1}{2\pi}\int_{-\pi}^{\pi} \mathrm{d}\varphi\,\sin\theta_\alpha(\varphi) = \langle\sin\theta_\alpha(\varphi)\rangle \tag{2.12}$$

and

$$A_n = \frac{1}{\pi}\int_{-\pi}^{\pi} \mathrm{d}\varphi\,\sin\theta_\alpha(\varphi)\cos(n\varphi), \quad n \geq 1 \tag{2.13}$$

leading to

$$F_z = 2\gamma\,\pi r_0 A_0. \tag{2.14}$$

The pressure force from the tilted surface is given by

$$N = \int_{-r_0}^{r_0} p(x)\,\mathrm{d}A, \quad \mathrm{d}A = 2\sqrt{r_0^2 - x^2}\,\mathrm{d}x \tag{2.15}$$

and

$$N = \pi p_c r_0^2. \tag{2.16}$$

The zero of pressure is chosen as the atmospheric pressure. Balance of forces in the normal direction leads to

$$N = F_z + mg\cos\alpha \tag{2.17}$$

or

$$\pi p_c r_0^2 = \gamma 2\pi r_0 A_0 + \rho V g \cos\alpha. \tag{2.18}$$

The pressure $p_c$ can be written in terms of the pressure at point $o$ (intersection point of the $z$-axis and the surface of the droplet), as follows:

$$p_c = p_o + \rho g h \cos\alpha \tag{2.19}$$

with $h$ the $oc$ height (figure 1). This pressure, with the use of the Young–Laplace equation, can also be written in terms of the mean-curvature $H_0$ at point $o$

$$p_o = -2\gamma H_0 \tag{2.20}$$

with $\gamma$ as the surface tension of liquid. Altogether the force balance (2.18) reads

$$\pi r_0^2(-2\gamma H_0 + \rho g h \cos\alpha) = 2\gamma\pi r_0 A_0 + \rho V g \cos\alpha \tag{2.21}$$

or, after dividing by $2\pi r_0^2 \gamma$,

$$-H_0 + \frac{\rho g h}{2\gamma}\cos\alpha = \frac{A_0}{r_0} + \frac{\rho V g}{2\pi r_0^2 \gamma}\cos\alpha. \tag{2.22}$$

The above is a direct generalization of eqn (12) of [14,15,19] for a horizontal substrate ($\alpha = 0$).

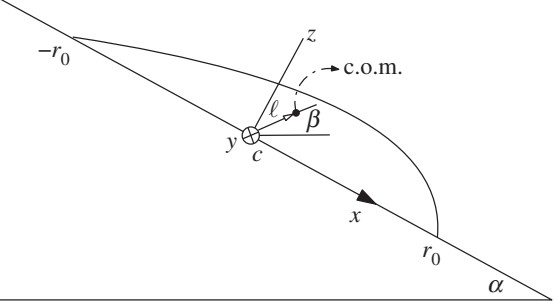

**Figure 2.** The geometry used for torque-balance identity.

It is convenient to have, with the help of the zero-gravity copy (spherical cap) of the droplet, a dimensionless form of the above identity. Using the spherical-cap radius $R_0$ and contact-angle $\theta_0$ with

$$r_0 = R_0 \sin \theta_0, \quad V = \frac{\pi R_0^3}{3}(1 - \cos \theta_0)^2(2 + \cos \theta_0) \tag{2.23}$$

one defines the dimensionless quantities [3]

$$\tilde{h} = \frac{h}{R_0}, \quad \tilde{H}_0 = R_0 H_0. \tag{2.24}$$

The dimensionless version of (2.22) is

$$- \tilde{H}_0 + \frac{3\tilde{h} \sin \theta_0 \, \mathrm{Bo} \cos \alpha}{\pi(1 - \cos \theta_0)^2(2 + \cos \theta_0)} = \frac{A_0}{\sin \theta_0} + \frac{\mathrm{Bo} \cos \alpha}{\pi \sin \theta_0}. \tag{2.25}$$

In §5, in order to confront theoretical and simulation values, we will evaluate and compare to 1 the ratio

$$\mathrm{ratio}_\perp = \frac{-\tilde{H}_0 \sin \theta_0 + \dfrac{3\tilde{h} \sin^2 \theta_0}{\pi(1 - \cos \theta_0)^2(2 + \cos \theta_0)} \mathrm{Bo} \cos \alpha}{A_0 + \dfrac{1}{\pi} \mathrm{Bo} \cos \alpha}. \tag{2.26}$$

# 3. Identity by torque balance

The contribution of the weight to the torque in the $y$-direction (inward figure 2) is given by

$$\tau_{wy} = mg\ell \cos \beta = \rho V g\ell \cos \beta \tag{3.1}$$

in which $\ell$ is the distance of the centre-of-mass (c.o.m.) of the droplet from the centre $c$, and $\beta$ is the angle between the c.o.m. position vector and the horizontal direction (figure 2). The torque applied by the substrate upon the drop can be evaluated by integration over the element by the pressure

$$\mathrm{d}\tau_{py} = -x\, p(x)\, \mathrm{d}A, \quad \mathrm{d}A = 2\sqrt{r_0^2 - x^2}\, \mathrm{d}x \tag{3.2}$$

leading to

$$\tau_{py} = -2 \int_{-r_0}^{r_0} (p_c + \rho g x \sin \alpha) x \sqrt{r_0^2 - x^2}\, \mathrm{d}x \tag{3.3}$$

and

$$\tau_{py} = -\frac{\pi}{4} \rho g r_0^4 \sin \alpha. \tag{3.4}$$

The infinitesimal capillary force at polar angle $\varphi$ with contact-angle $\theta_\alpha(\varphi)$ reads

$$\mathrm{d}\boldsymbol{F}_\gamma = \gamma r_0\, \mathrm{d}\varphi \Big( \cos \theta_\alpha(\varphi) \cos \varphi\, \mathbf{i} + \cos \theta_\alpha(\varphi) \sin \varphi\, \mathbf{j} - \sin \theta_\alpha(\varphi)\, \mathbf{k} \Big) \tag{3.5}$$

with the position vector

$$\boldsymbol{r} = r_0 \Big( \cos \varphi\, \mathbf{i} + \sin \varphi\, \mathbf{j} \Big) \tag{3.6}$$

leading to the torque element in $y$-direction

$$\mathrm{d}\tau_{\gamma y} = (r \times \mathrm{d}F_\gamma)_y = \gamma r_0^2 \sin \theta_\alpha(\varphi) \cos \varphi \, \mathrm{d}\varphi. \tag{3.7}$$

The integration over the above torque element gives

$$\tau_{\gamma y} = \gamma r_0^2 \int_{-\pi}^{\pi} \sin \theta_\alpha(\varphi) \cos \varphi \, \mathrm{d}\varphi. \tag{3.8}$$

Again using Fourier expansion (2.11) for $\sin \theta_\alpha(\varphi)$ one finds

$$\tau_{\gamma y} = \pi \gamma r_0^2 A_1. \tag{3.9}$$

All together, the balance of torques along $y$-direction gives

$$\rho V g \ell \cos \beta + \pi \gamma r_0^2 A_1 = \frac{\pi}{4} \rho g r_0^4 \sin \alpha. \tag{3.10}$$

Again it is convenient to have a dimensionless form of the identity. Defining

$$\tilde{\ell} = \frac{\ell}{R_0} \tag{3.11}$$

and using (2.23) and the relation between two Bond numbers [8]

$$\mathrm{Bo} = \pi B \frac{(1 - \cos \theta_0)^2 (2 + \cos \theta_0)}{6 \sin \theta_0} \tag{3.12}$$

the dimensionless form of identity (3.10) reads

$$2 \,\mathrm{Bo}\,\tilde{\ell} \cos \beta + \pi \sin \theta_0 A_1 = \frac{\pi}{4} B \sin^3 \theta_0 \sin \alpha. \tag{3.13}$$

In §5, the following ratio is compared to 1 by the numerical simulations

$$\mathrm{ratio}_{\circlearrowleft} = \frac{2 \,\mathrm{Bo}\,\tilde{\ell} \cos \beta + \pi \sin \theta_0 A_1}{\dfrac{\pi}{4} B \sin^3 \theta_0 \sin \alpha}. \tag{3.14}$$

# 4. Linear response

## 4.1. Check at linear response, along surface

The linear response ansatz [3] is in terms of the Bond number

$$B = \frac{\rho g R_0^2}{\gamma}, \tag{4.1}$$

where $R_0$ is the radius of the spherical cap at zero gravity, see (2.23). In the linear response approximation we have [3,8]

$$C_1 = -\frac{1}{2} \left( \cos \theta_\alpha^{\min} - \cos \theta_\alpha^{\max} \right) + O(B^2) \tag{4.2}$$

in which $\theta_\alpha^{\max} = \theta_\alpha(0)$ and $\theta_\alpha^{\min} = \theta_\alpha(\pi)$. We see that the expression (2.7) in linear approximation is the famous Furmidge relation (eqn 1 of [9]), with the constant $K = \pi/4$.

The check of (2.8) by explicit expressions for $\cos \theta_\alpha^{\min}$ and $\cos \theta_\alpha^{\max}$ at the linear approximation [8] is straightforward, once the relation between two Bond numbers (3.12) being used.

## 4.2. Check at linear response, normal to surface

The linear response ansatz [3,8] implies

$$\cos \theta_\alpha(\varphi) = \cos \theta_0 + \lambda B + \mu B \cos \varphi + O(B^2), \tag{4.3}$$

where $\theta_0$ is the uniform contact angle at $B = 0$, $\lambda$ and $\mu$ are some constants whose values do not

matter here. Then

$$\sin \theta_\alpha(\varphi) = \sqrt{1 - \cos^2 \theta_\alpha(\varphi)}$$

$$= \sin \theta_0 - \lambda B \cos \theta_0 / \sin \theta_0 - \mu B \cos \varphi \cos \theta_0 / \sin \theta_0 + O(B^2) \tag{4.4}$$

which implies both the average over $\varphi$

$$A_0 = \langle \sin \theta_\alpha(\varphi) \rangle = \sin \theta_0 - \lambda B \cot \theta_0 + O(B^2) \tag{4.5}$$

and the arithmetic mean between maximum and minimum

$$\frac{1}{2}(\sin \theta_\alpha^{\max} + \sin \theta_\alpha^{\min}) = \sin \theta_0 - \lambda B \cot \theta_0 + O(B^2). \tag{4.6}$$

Hence

$$A_0 = \frac{1}{2}\left(\sin \theta_\alpha^{\min} + \sin \theta_\alpha^{\max}\right) + O(B^2). \tag{4.7}$$

We have $\tilde{h} = 1 - \cos \theta_0 + O(B)$ [3]. By the above, identity (2.25) comes to the form

$$-\tilde{H}_0 = \frac{\sin \theta_\alpha^{\min} + \sin \theta_\alpha^{\max}}{2 \sin \theta_0} - \frac{B}{6}\frac{(1 - \cos \theta_0)(1 + 2\cos \theta_0)}{1 + \cos \theta_0}\cos \alpha + O(B^2). \tag{4.8}$$

The $\tilde{H}_0$, $\sin \theta_\alpha^{\min}$ and $\sin \theta_\alpha^{\max}$ are read from [3,8], by which we have

$$-\tilde{H}_0 = 1 - \frac{B}{6}(1 - \cos \theta_0)\cos \alpha + O(B^2) \tag{4.9}$$

and

$$\frac{1}{2}(\sin \theta_\alpha^{\min} + \sin \theta_\alpha^{\max}) = \sin \theta_0 - B \cos \theta_0 r'_{01}(\theta_0)\cos \alpha + O(B^2) \tag{4.10}$$

with [3]

$$r'_{01}(\theta_0) = -\frac{\sin \theta_0}{6} + \frac{\sin \theta_0 \cos \theta_0}{3(1 + \cos \theta_0)}. \tag{4.11}$$

It is a simple matter to check that the quantities (4.7)–(4.10) inserted into (2.25) satisfy (2.25) up to $O(B^2)$.

## 4.3. Check at linear response: torque

Proceeding as in §4.2, the linear approximation yields

$$A_1 = -\frac{1}{2}\left(\sin \theta_\alpha^{\min} - \sin \theta_\alpha^{\max}\right) + O(B^2) \tag{4.12}$$

$$= \frac{1}{3}B\frac{(1 - \cos \theta_0)(2 + \cos \theta_0)}{1 + \cos \theta_0}\cos \theta_0 \sin \alpha + O(B^2). \tag{4.13}$$

For a check at linear order, it is enough to insert the spherical cap droplet values in the terms having $B$ or Bo, namely the first and last terms in (3.13). The c.o.m. of a spherical cap is known. Subtracting $R_0 \cos \theta_0$ leads to

$$\tilde{\ell} = \frac{3(1 + \cos \theta_0)^2}{4(2 + \cos \theta_0)} - \cos \theta_0 + O(B). \tag{4.14}$$

Also for the spherical cap the c.o.m. lays on the $z$-axis, for which we have $\beta = \pi/2 - \alpha$. By the relation between the two Bond numbers (3.12), it is easy to see that identity (3.13) is satisfied up to $O(B^2)$.

# 5. Numerical check of identities

To test the identities (2.8) and (2.25), the solutions of the Young–Laplace equation are developed by the *Surface Evolver* software, at different slope angles $\alpha$s, spherical cap contact angles $\theta_0$s and modified Bond numbers Bo $= mg/(2r_0\gamma)$. The identities take the form

$$\text{ratio}_\parallel = 1, \quad \text{ratio}_\perp = 1, \quad \text{ratio}_\circlearrowleft = 1$$

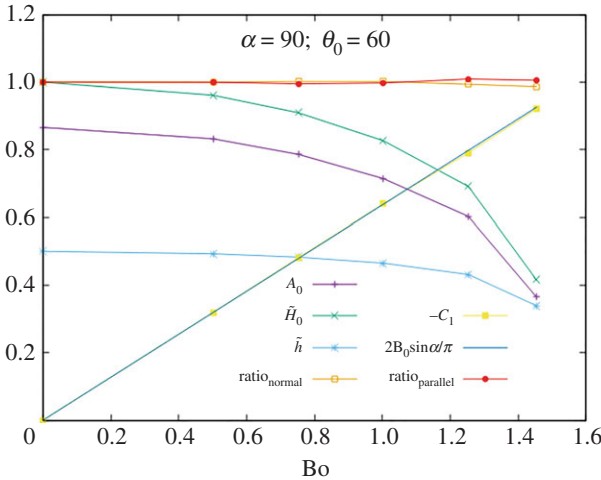

**Figure 3.** Testing the identities for $\alpha = 90°$ and $\theta_0 = 60°$.

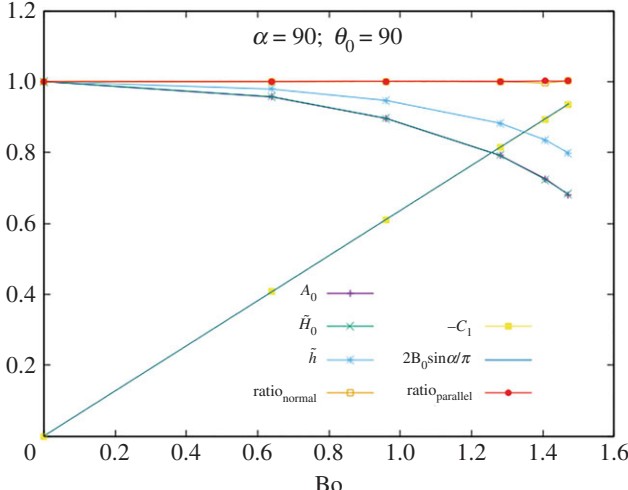

**Figure 4.** Testing the identities for $\alpha = 90°$ and $\theta_0 = 90°$.

with the ratios defined in (2.9), (2.26) and (3.14). In order to measure the mean curvature $\tilde{H}_0$ and height $\tilde{h}$ above the origin, we export from *Surface Evolver* the list of vertices such that in cylindrical coordinates $\tilde{r} < 0.06\,\tilde{r}_0$, that is 6 per cent of the contact radius. This yields between 80 and 110 vertices. We then fit a quadratic surface

$$\tilde{z} = a\tilde{x}^2 + b\tilde{y}^2 + c\tilde{x} + \tilde{h} \tag{5.1}$$

to obtain $\tilde{h}$ and

$$-\tilde{H}_0 = (a + b(1 + c^2))(1 + c^2)^{-3/2} + O(B^2). \tag{5.2}$$

The measurement of $A_0$ and $C_1$ implies more work, described in §6.

In figures 3–5 we display $A_0$, $\tilde{H}_0$, $\tilde{h}$, ratio$_\perp$, $-C_1$ and ratio$_\parallel$ as functions of Bo for $\alpha = 90°$ and $\theta_0 = 60°$, 90°, 120°. Tests were made with the following values:

$$\left.\begin{array}{l} \alpha = 45°,\, 90°,\, 135° \\ \theta_0 = 60°,\, 90°,\, 120° \end{array}\right\} \tag{5.3}$$

and many different values of the Bo number, away from the singularity, giving consistent values in all cases. As a sample of the numerical values, the data for the case with $\alpha = 90°$ and $\theta_0 = 90°$ are given in table 1.

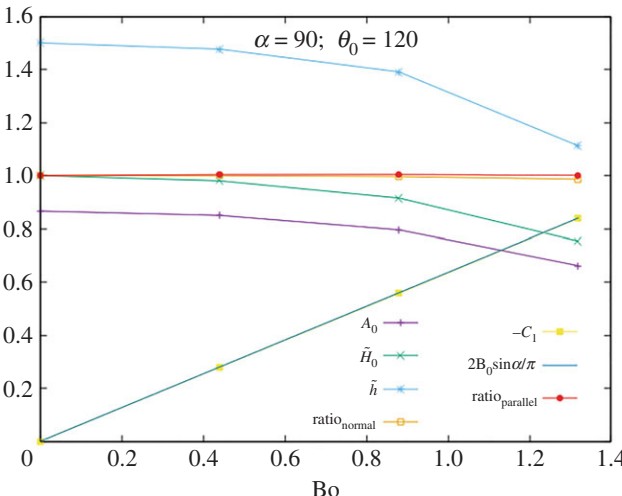

**Figure 5.** Testing the identities for $\alpha = 90°$ and $\theta_0 = 120°$.

**Table 1.** Testing the identities for $\alpha = 90°$ and $\theta_0 = 90°$.

| Bo | ratio$_{\parallel}$ | ratio$_{\perp}$ |
|---|---|---|
| 0.63971 | 1.0000 | 1.0006 |
| 0.95957 | 1.0010 | 1.0001 |
| 1.2794 | 1.0008 | 0.99987 |
| 1.4073 | 1.0023 | 0.99609 |
| 1.4713 | 1.0017 | 1.0020 |

In order to check the torque-balance identity numerically it is needed to find the c.o.m. of the droplet, for which out of the irregular distribution of vertices one has to extract regular d$x$ × d$y$-mesh and d$y$ × d$z$-mesh, for calculating $\tilde{z}_{cm}$ and $\tilde{x}_{cm}$, respectively. The result of making meshes for one of the samples is given in figures 6 and 7. The values of $\tilde{l}$ and angle $\beta$ are then obtained by

$$\beta = \tan^{-1}\frac{z_{cm}}{x_{cm}} - \alpha \tag{5.4}$$

and

$$\tilde{\ell} = \sqrt{\tilde{x}_{cm}^2 + \tilde{z}_{cm}^2}. \tag{5.5}$$

To obtain the average-value $A_1$ the azimuth-angle is needed; the computation is postponed to §6. The results of torque identity checks are summarized in table 2.

# 6. Contact angle as function of azimuth

In order to check the *Surface Evolver* simulations against the exact identities, we have to compute $C_0$ and $C_1$, or the average over $\varphi \in (0, 2\pi)$ of $\sin\theta(\varphi)$ and the average over $\varphi \in (0, 2\pi)$ of $\cos\varphi\cos\theta(\varphi)$, where $\theta(\varphi)$ is the contact angle at azimuth $\varphi$. A fluid interface in contact with a solid surface has a contact angle $\theta \in (0, \pi)$ which may vary along the contact line. The solid surface must be smooth at the macroscopic scale, so that a unique normal vector is defined at every point.

We have performed the *Surface Evolver* simulations with Dirichlet boundary conditions: the displacement is zero on the contact line, a fixed circle of radius $r_0$. Physically, other than a droplet on or below an incline, it may represent a pocket of liquid made with an elastic membrane fixed onto a circular metallic wire, opening to a reservoir of liquid. Initially, at zero gravity, the pocket has the shape of a spherical cap of contact angle $\theta_0$, giving the desired volume. The pocket lies entirely on one side, say $\{z \geq 0\}$, of the plane containing the contact line circle. Upon switching on the gravity, the pocket deforms

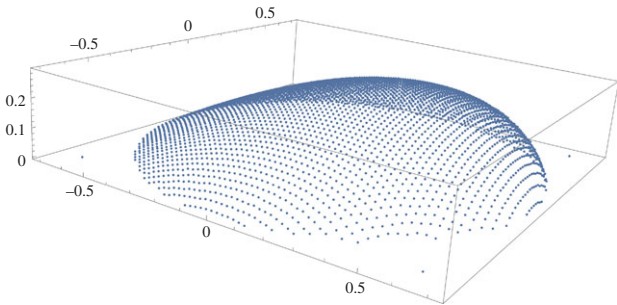

**Figure 6.** $dx \times dy$ mesh for sample $\alpha = 30°$ and $\theta_0 = 45°$, with Bo $= 1.015$. Mesh size: $0.01 \times 0.01$.

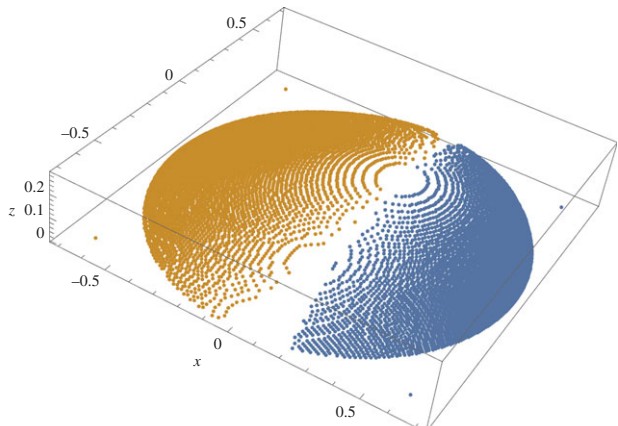

**Figure 7.** $dy \times dz$ mesh for the sample of figure 6. Mesh size: $0.01 \times 0.005$. Points $x > x_{zmax}$ in blue and $x < x_{zmax}$ in orange.

**Table 2.** Testing the torque-balance identity.

| $\alpha$ | $\theta_0$ | Bo | ratio |
| --- | --- | --- | --- |
| 30° | 45° | 1.02 | 0.999 |
| 30° | 90° | 0.512 | 1.01 |
| 45° | 30° | 0.453 | 0.999 |
| 60° | 60° | 0.401 | 1.00 |

as a solution of the Young–Laplace equation under constant volume constraint and the given Dirichlet boundary conditions. Eventually the interface or membrane may go partly into $\{z < 0\}$ region.

Of course, so far as the interface remains in $\{z \geq 0\}$, it represents as well a liquid drop on or below a solid plane substrate, with contact line pinned on the circle and contact angle $\theta \in (0, \pi)$ varying along the circle. However, where the interface goes into $\{z < 0\}$, representing a pocket, a contact angle $\theta > \pi$ or $\theta < 0$ will be found. This occurs e.g. for $\alpha = \pi/2$, $\theta_0 = \pi/3$, Bo $= 1.46$ around $\varphi = \pi$ (figure 8) and for $\alpha = \pi/2$, $\theta_0 = 2\pi/3$, Bo $= 1.32$ around $\varphi = 0$ (figure 9). At such parameters, a droplet on the incline would have unpinned, dewetting from the top around $\varphi = \pi$ or overflowing at the bottom around $\varphi = 0$. The Young–Laplace equation then has to be solved with moving boundary, which we have not done.

The figures also show the linear response approximation, namely $\cos\theta$ as a linear function of $\cos\varphi$, at the smallest Bond number for which data are displayed.

As another noticeable feature, it is observed that to some approximation, for each choice of $\alpha$ and $\theta_0$, there is an azimuth $\varphi$ at which $\theta \simeq \theta_0 \ \forall$Bo.

The number of mesh vertices in the *Surface Evolver* simulation is up to 700 000, with finer mesh near the contact line and finer mesh also in the more delicate cases where the Bond number Bo approaches the instability threshold. In order to measure the contact angle $\theta$ as function of the azimuth $\varphi$, we export from *Surface Evolver* the list of vertices such that $|\tilde{z}| < 0.03\,(1 - \cos\theta_0)$, that is 3 per cent of the height of the drop at zero Bond number. This yields between 1300 and 20 000 vertices. We divide the range of

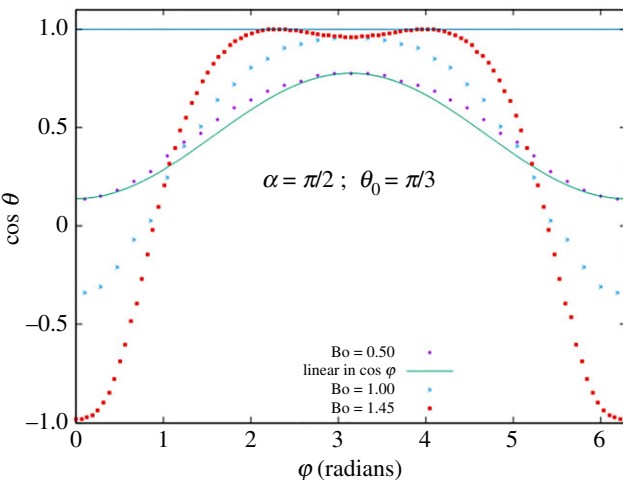

**Figure 8.** Cosine of contact angle $\theta$ as function of azimuth $\varphi$ for $\alpha = 90°$ and $\theta_0 = 60°$ at Bond numbers Bo = 0.50, 1.00, 1.45. Linear response approximation is plotted at Bo = 0.50.

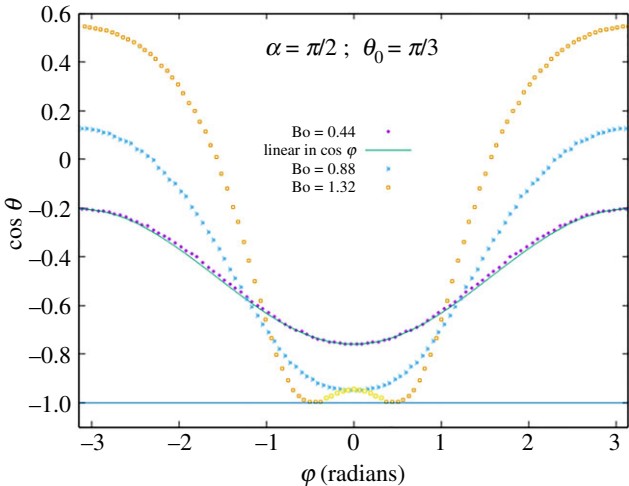

**Figure 9.** Cosine of contact angle $\theta$ as function of azimuth $\varphi$ for $\alpha = 90°$ and $\theta_0 = 120°$ at Bond numbers Bo = 0.44, 0.88, 1.32. Linear response approximation is plotted at Bo = 0.44.

azimuth into a hundred equal sectors. Each sector contains between four and six hundred vertices. In each sector we fit with *gnuplot* a surface containing the contact line in the form, in cylindrical coordinates,

$$r(z) = r_0 - z \cot \theta + az^2 \quad \text{or} \quad z(r) = -(r - r_0) \tan \theta + b(r - r_0)^2.$$

The fit is performed, independently in each sector, in terms of $\cot \theta$ and $a$ or $\tan\theta$ and $b$.

# 7. Uses of identities

## 7.1. In first place: an illustrative example

The identity by force balance in normal direction can be used in the first place in the Young–Laplace equation. It helps to replace some unknown values in the equation from the beginning. This has been used in [14] to develop a perturbation solution and in [19] to reduce the numerical procedure to a simple shooting method, for sessile drops on horizontal surface with fixed contact-angle. Here we show that the identity can be used to replace the unknown quantities in the Young–Laplace equation for drops with fixed contact radius (pinned drops) on tilted surfaces as well.

First one recognizes that the Young–Laplace equation can be written in the form

$$-2\gamma H_0 + \rho g(h - z)\cos \alpha = -2\gamma H \tag{7.1}$$

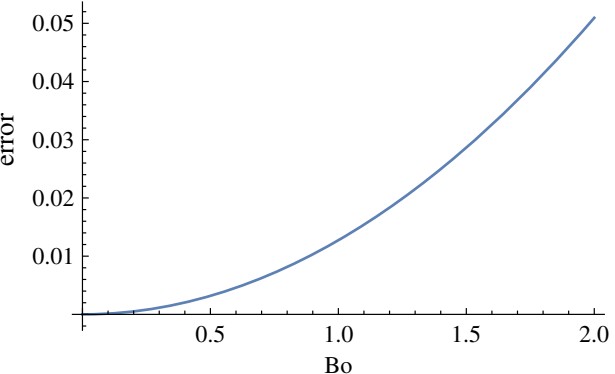

**Figure 10.** The estimation of error of $-\tilde{H}_0$ versus Bo for $\alpha = 45°$ and $\theta_0 = 120°$.

in which $H_0$ and $h$ are unknown in the first place. Now by the identity (2.22) one has for the combination as follows:

$$-2\gamma H_0 + \rho g h \cos \alpha = 2\gamma \frac{A_0}{r_0} + \frac{\rho V g}{\pi r_0^2} \cos \alpha \tag{7.2}$$

bringing the Young–Laplace to the form

$$2\gamma \frac{A_0}{r_0} + \frac{\rho V g}{\pi r_0^2} \cos \alpha - \rho g z \cos \alpha = -2\gamma H. \tag{7.3}$$

Now in the r.h.s. only $A_0$ is unknown. To develop a perturbative analytical solution, such as in [3], the above form has the advantage that it is involved by less to be calculated values. At linear order of $B$ number, the above form finds its full advantage, because at the beginning even $A_0$ is known by the linear ansatz (4.7). As mentioned earlier, the numerical solution in reduced form helps lowering the parameters space dimension [19].

## 7.2. Estimation of error by identities

As mentioned earlier, the exact mathematical relation may be used to make an estimation error for an analytical or numerical approach. As an example, we use identity (2.25) by force balance normal to the substrate to estimate the error of $-\tilde{H}_0$ by the linear approximation of [8], that is expected to be of second order in $B$. First let us rearrange the identity in the form that the apex curvature would be calculated by other values, namely

$$[-\tilde{H}_0]_{\text{Identity}} = -\frac{3\tilde{h} \sin \theta_0 \, \text{Bo} \cos \alpha}{\pi (1 - \cos \theta_0)^2 (2 + \cos \theta_0)} + \frac{A_0}{\sin \theta_0} + \frac{\text{Bo} \cos \alpha}{\pi \sin \theta_0}. \tag{7.4}$$

The above gives the value of $-\tilde{H}_0$ required to hold the force-balance condition. It was seen earlier in §4.2 that, when only the first order is kept the above equality holds. Now

$$\tilde{h} = 1 - \cos \theta_0 + \frac{B}{6} \left( 1 - \cos \theta_0 + 2 \ln \frac{1 + \cos \theta_0}{2} \right) \cos \alpha + O(B^2). \tag{7.5}$$

When expressed based on the number Bo via relation (3.12) (also (8) of [8]), the estimation of error is given by:

$$\frac{[-\tilde{H}_0]_{\text{identity}} - [-\tilde{H}_0]_{\text{linear sol}}}{\tilde{R}_0^{-1}} = \frac{-\text{Bo}^2}{3\pi^2} \frac{\sin^2 \theta_0}{(1 - \cos \theta_0)^4 (2 + \cos \theta_0)^2}$$
$$\times \left( 1 - \cos \theta_0 + 2 \ln \frac{1 + \cos \theta_0}{2} \right) \cos^2 \alpha \tag{7.6}$$

in which we have used $\tilde{R}_0 = 1$. The remarkable observation by [8] is that, for relatively large values of the Bond number Bo the linear approximation solution is quite close to the numerical solution. In figure 10, we see that for Bond numbers up to Bo = 2.0 the error is less than 5%.

# 8. Conclusion

The main concern of the present work is to highlight exact mathematical identities for droplets, even in cases where an exact Young–Laplace profile solution is not available. As a special case, droplets with circular contact boundary based on flat tilted surfaces are considered. Two of the identities are derived as the requirements of the force balance, and one of the torque balance. The identities involve some relevant values of droplets. On the side of output values, some are available once an analytical or numerical solution would be given, for instance the curvature at apex or the vertical height. Some of these output values appearing in the identities are somehow indirectly available, such as certain azimuthal-angle averages, or the location of the centre-of-mass of the droplet. The identities are put under test both by the available solutions of a linear response approximation as well as the ones obtained from exact numerical solutions. We stress that the problem analysed here should not only be seen as an abstract theoretical construction. It is a real problem having an experimental counterpart, e.g. putting the droplet on a disc-shaped asperity on the substrate.

Data accessibility. This article has no additional data.
Authors' contributions. F.D. participated to define the problem, developed numerical analysis, drafted parts of text; A.H.F. suggested the study, contributed to maths derivations, drafted parts of text; M.H. contributed to maths derivations and fixing the results, critical reading the manuscript; T.H. participated in shaping the problem, contributed in checking and completing maths expressions, drafting parts of text and making final form of paper. All authors gave final approval for publication and agree to be held accountable for the work performed therein.
Competing interests. We declare we have no competing interests.
Funding. The work by A.H.F. is supported by the Research Council of Alzahra University.

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
