## [Reviewer comments · Royal Society Open Science]

Review History

RSOS-201534.R0 (Original submission)

Review form: Reviewer 1

Is the manuscript scientifically sound in its present form?

Yes

Are the interpretations and conclusions justified by the results?

Yes

Is the language acceptable?

Yes

Do you have any ethical concerns with this paper?

Yes

Have you any concerns about statistical analyses in this paper?

No

Recommendation?

Major revision is needed (please make suggestions in comments)

Comments to the Author(s)

This work concerns the static shape of pinned (on a circular basis) droplets for a flat substrate undergoing tilting. This shape can be described by the Young-Laplace equation. Three formal integral equilibrium relations are derived. The interesting issue is the insertion of the polar contact angle distribution in these. The validity of the relations is checked through small Bo perturbation analysis and by comparing with numerical solution (via Surface Evolver). The significance of the integral relations in derivation of approximate solutions and on checking numerical solution is discussed.

Overall the work is sophisticated, elegant and useful. However there is a need for improvement.

-It is surprisingly that even today the simple YL equation is a research subject. Recently professor Runckestein published two papers in Langmuir discussing how the Bond number must be defined (seemingly a trivial issue). I recommend this papers to the authors.

- The circular shape of the contact line must appear in the title (the term "pinned" alone is too general).

- I understand that "pendent" may be grammatically better but I remind you that the literature use the word "pendant" for the particular problem.

-The use of the expression "input and output parameters" although is conceivable to me, I think it will be confused for the general readers. After all, the distinction between the parameters depends on the problem at hand. It may be replaced by "relating the problem parameters" or something like that.

- Line 51 of page 2. This issue has also been study (semi-analytically) in

Kostoglou, M., Karapantsios, T. Contact angle profiles for droplets on omniphilic surfaces in the presence of tangential forces. *Colloids Interfaces* 3, 60, 2019.

for slender unpinned droplets with evolving contact line shape based on experimentally observed contact lines in

Ríos-López, I., Evgenidis, S., Kostoglou, M., Zabulis, X., Karapantsios, T.D. "Effect of initial droplet shape on the tangential force required for spreading and sliding along a solid surface" *Colloids and Surfaces A: Physicochemical and Engineering Aspects* 549, 164-173, 2018.

The above must be discussed by the authors.

-I do not think that the description of Surface Evolver as finite element method (which has a very specific meaning in numerical analysis) is appropriate.

-Use "dimensionless" instead of "a dimensional. In general there is a need for improvement in the syntax and language.

-I noticed that there is no a verification ratio for the identity 3 as for the other two. Also there is no validation for this identity in comparing with numerical solution.

-I would prefer to see actual contact angles (no cosines) in figs 6 and 7. Is the bimodality appeared physically correct or it is a numerical error?

- The lines 45-55 are quite inappropriate for the conclusions section and must be removed.

-Finally the problem analyzed should not be seen as an abstractive theoretical construction. It is a real problem having an experimental counterpart (e.g. putting the droplet on a disc shaped asperity (step) on the substrate). This must be noticed.

Review form: Reviewer 2

Is the manuscript scientifically sound in its present form?

Yes

Are the interpretations and conclusions justified by the results?

Yes

Is the language acceptable?

Yes

Do you have any ethical concerns with this paper?

No

Have you any concerns about statistical analyses in this paper?

No

Recommendation?

Accept with minor revision (please list in comments)

Comments to the Author(s)

The authors present three identities pertaining a drop resting on an inclined surface. The identities stem from the fact that the net force and torque acting on a drop at rest must be zero. The authors not only provide a mathematical derivation of such identities, they also provide numerical and analytic (perturbative) checks. I find the paper novel and very interesting. I also find the paper well written and easy to follow. I have just a few questions that I would like the authors to answer.

(1) My main question relates to the roles played by hydrostatic pressure and the force N of Eq. (15). Hydrostatic pressure is usually understood as coming from the weight of the liquid above a point. Hence, when the authors calculate the force N in Eq. (15), shouldn't that force be given by the weight of the drop, since $p(x)$ should just be due to the weight of the liquid above point x ?

(2) Related to (1): In analogy to the solid-solid case, one may think that there actually should exist a force N , but that force should be the normal force (i.e., the reaction force of the surface on the drop). That normal force N should be determined from Eq. (17), but it should be not known a priori. Why is not there a normal force acting on the drop that is analogous to the solid-solid normal force?

(3) Related to (1) and (2): For a pendant drop that is hanging from the surface, there should be no hydrostatic pressure acting on the surface, and hence the weight is equal to the capillary force, that is, N should be zero in the pendant case. Is that so?

(4) Shouldn't the effect of the Laplace pressure be included in a similar way to the hydrostatic pressure? If the answer is yes, would that lead to another normal force that should be added to the N of Eq. (16)? If the answer is no, why not?

(5) Their numerical simulations are very nice. However, I have a minor concern. The triple line of many common liquids (e.g., water) at the angle of inclinations used in the simulations (45, 90, and 135) would not remain circumferential. Do the authors know if their results change drastically if the triple line is not circumferential? (I know the authors mention in the conclusions that this is an open problem, I am just curious to know if they have any numerical results on this.)

(6) Finally, a minor thing. The authors use the coefficients $C1$, $A0$ and $A1$ that are part of two Fourier expansions. However, it seems to me that their results could be obtained without ever making any reference to such Fourier expansions. Is that so? If not, do the other coefficients of such Fourier expansions play any role in these (or any other) drop identities?

Decision letter (RSOS-201534.R0)

Dear Dr Fatollahi

On behalf of the Editors, we are pleased to inform you that your Manuscript RSOS-201534 "Identities for Pinned Droplets on Tilted Surfaces" has been accepted for publication in Royal Society Open Science subject to minor revision in accordance with the referees' reports. Please find the referees' comments along with any feedback from the Editors below my signature.

Please submit your revised manuscript and required files (see below) no later than 7 days from today's (ie 30-Sep-2020) date. Note: the ScholarOne system will 'lock' if submission of the revision is attempted 7 or more days after the deadline. If you do not think you will be able to meet this deadline please contact the editorial office immediately.

Best regards,

on behalf of Professor Guy Genin (Associate Editor) and Miles Padgett (Subject Editor)
openscience@royalsociety.org

Associate Editor Comments to Author (Professor Guy Genin):

Many thanks to you for submitting this very nice contribution to Royal Society Open Science. Like the reviewers, I am enthusiastic about your work. I believe that this is an important paper that will advance the field.

I also agree with the reviewers that the manuscript does require minor but substantial work prior to publication. Although the importance of the manuscript is clear, the reviewers have a range of constructive comments that I believe will elevate the paper. Please do address the comments of the reviewers in your revised manuscript.

I am looking forward to reading your revised work!

Guy Genin

Reviewer comments to Author:

Reviewer: 1

Comments to the Author(s)

This work concerns the static shape of pinned (on a circular basis) droplets for a flat substrate undergoing tilting. This shape can be described by the Young-Laplace equation. Three formal integral equilibrium relations are derived. The interesting issue is the insertion of the polar contact angle distribution in these. The validity of the relations is checked through small Bo perturbation analysis and by comparing with numerical solution (via Surface Evolver). The significance of the integral relations in derivation of approximate solutions and on checking numerical solution is discussed.

Overall the work is sophisticated, elegant and useful. However there is a need for improvement.

-It is surprisingly that even today the simple YL equation is a research subject. Recently professor Runckestein published two papers in Langmuir discussing how the Bond number must be defined (seemingly a trivial issue). I recommend this papers to the authors.

- The circular shape of the contact line must appear in the title (the term "pinned" alone is too general).

- I understand that "pendent" may be grammatically better but I remind you that the literature use the word "pendant" for the particular problem.

-The use of the expression "input and output parameters" although is conceivable to me, I think it will be confused for the general readers. After all, the distinction between the parameters depends on the problem at hand. It may be replaced by "relating the problem parameters" or something like that.

- Line 51 of page 2. This issue has also been study (semi-analytically) in

Kostoglou, M., Karapantsios, T. Contact angle profiles for droplets on omniphilic surfaces in the presence of tangential forces. *Colloids Interfaces* 3, 60, 2019.

for slender unpinned droplets with evolving contact line shape based on experimentally observed contact lines in

Ríos-López, I., Evgenidis, S., Kostoglou, M., Zabulis, X., Karapantsios, T.D. "Effect of initial droplet shape on the tangential force required for spreading and sliding along a solid surface" *Colloids and Surfaces A: Physicochemical and Engineering Aspects* 549, 164-173, 2018.

The above must be discussed by the authors.

-I do not think that the description of Surface Evolver as finite element method (which has a very specific meaning in numerical analysis) is appropriate.

-Use "dimensionless" instead of "a dimensional. In general there is a need for improvement in the syntax and language.

-I noticed that there is no a verification ratio for the identity 3 as for the other two. Also there is no validation for this identity in comparing with numerical solution.

-I would prefer to see actual contact angles (no cosines) in figs 6 and 7. Is the bimodality appeared physically correct or it is a numerical error?

- The lines 45-55 are quite inappropriate for the conclusions section and must be removed.

-Finally the problem analyzed should not be seen as an abstractive theoretical construction. It is a real problem having an experimental counterpart (e.g. putting the droplet on a disc shaped asperity (step) on the substrate). This must be noticed.

Reviewer: 2

Comments to the Author(s)

The authors present three identities pertaining a drop resting on an inclined surface.

The identities stem from the fact that the net force and torque acting on a drop at rest

must be zero. The authors not only provide a mathematical derivation of such identities, they also provide numerical and analytic (perturbative) checks. I find the paper novel and very interesting.

I also find the paper well written and easy to follow.

I have just a few questions that I would like the authors to answer.

(1) My main question relates to the roles played by hydrostatic pressure and the force N of Eq. (15). Hydrostatic pressure is usually understood as coming from the weight of the liquid above a point. Hence, when the authors calculate the force N in Eq. (15), shouldn't that force be given by the weight of the drop, since $p(x)$ should just be due to the weight of the liquid above point x ?

(2) Related to (1): In analogy to the solid-solid case, one may think that there actually should exist a force N , but that force should be the normal force (i.e., the reaction force of the surface on the drop). That normal force N should be determined from Eq. (17), but it should be not known a priori. Why is not there a normal force acting on the drop that is analogous to the solid-solid normal force?

(3) Related to (1) and (2): For a pendant drop that is hanging from the surface, there should be no hydrostatic pressure acting on the surface, and hence the weight is equal to the capillary force, that is, N should be zero in the pendant case. Is that so?

(4) Shouldn't the effect of the Laplace pressure be included in a similar way to the hydrostatic pressure? If the answer is yes, would that lead to another normal force that should be added to the N of Eq. (16)? If the answer is no, why not?

(5) Their numerical simulations are very nice. However, I have a minor concern. The triple line of many common liquids (e.g., water) at the angle of inclinations used in the simulations (45, 90, and 135) would not remain circumferential. Do the authors know if their results change drastically if the triple line is not circumferential? (I know the authors mention in the conclusions that this is an open problem, I am just curious to know if they have any numerical results on this.)

(6) Finally, a minor thing. The authors use the coefficients C_1 , A_0 and A_1 that are part of two Fourier expansions. However, it seems to me that their results could be obtained without ever making any reference to such Fourier expansions. Is that so? If not, do the other coefficients of such Fourier expansions play any role in these (or any other) drop identities?

===PREPARING YOUR MANUSCRIPT===

- one version identifying all the changes that have been made (for instance, in coloured highlight, in bold text, or tracked changes);
- a 'clean' version of the new manuscript that incorporates the changes made, but does not highlight them. This version will be used for typesetting.

If you have been asked to revise the written English in your submission as a condition of publication, you must do so, and you are expected to provide evidence that you have received language editing support. The journal would prefer that you use a professional language editing service and provide a certificate of editing, but a signed letter from a colleague who is a native speaker of English is acceptable. Note the journal has arranged a number of discounts for authors

using professional language editing services
(<https://royalsociety.org/journals/authors/benefits/language-editing/>).

===PREPARING YOUR REVISION IN SCHOLARONE===

-- If you have uploaded ESM files, please ensure you follow the guidance at <https://royalsociety.org/journals/authors/author-guidelines/#supplementary-material> to include a suitable title and informative caption. An example of appropriate titling and captioning may be found at https://figshare.com/articles/Table_S2_from_Is_there_a_trade-off_between_peak_performance_and_performance_breadth_across_temperatures_for_aerobic_sc_ope_in_teleost_fishes_/3843624.

Author's Response to Decision Letter for (RSOS-201534.R0)

See Appendix A.

RSOS-201534.R1 (Revision)

Review form: Reviewer 1

Is the manuscript scientifically sound in its present form?

Yes

Are the interpretations and conclusions justified by the results?

Yes

Is the language acceptable?

Yes

Do you have any ethical concerns with this paper?

Yes

Have you any concerns about statistical analyses in this paper?

No

Recommendation?

Accept as is

Comments to the Author(s)

The authors followed the suggestions proposed by the reviewers or answered appropriately to the comments so the manuscript is now suitable for publication.

Review form: Reviewer 2

Is the manuscript scientifically sound in its present form?

Yes

Are the interpretations and conclusions justified by the results?

Yes

Is the language acceptable?

Yes

Do you have any ethical concerns with this paper?

No

Have you any concerns about statistical analyses in this paper?

No

Recommendation?

Accept as is

Comments to the Author(s)

I recommend publication of the article.

Decision letter (RSOS-201534.R1)

Dear Dr Fatollahi,

It is a pleasure to accept your manuscript entitled "Identities for Droplets with Circular Footprint on Tilted Surfaces" in its current form for publication in Royal Society Open Science. The comments of the reviewer(s) who reviewed your manuscript are included at the foot of this letter.

on behalf of Professor Guy Genin (Associate Editor) and Miles Padgett (Subject Editor)
openscience@royalsociety.org

Associate Editor Comments to Author (Professor Guy Genin):

Congratulations on an outstanding contribution to the literature. Thank you for sending this excellent paper to RSOS!

Reviewer comments to Author:

Reviewer: 1

Comments to the Author(s)

The authors followed the suggestions proposed by the reviewers or answered appropriately to the comments so the manuscript is now suitable for publication.

Reviewer: 2

Comments to the Author(s)

I recommend publication of the article.

Appendix A

Amendments to the manuscript ID RSOS-201534:

"Identities for Pinned Droplets on Tilted Surfaces"

by: F. Dunlop, A.H. Fatollahi, M. Hajirahimi, and Th. Huillet

Dear Editors of Royal Society OS, dear Reviewers. Please find below the list of amendments (in **black**) made in response to our Reviewers queries.

- *Reviewer: 1*

Comments to the Author(s)

This work concerns the static shape of pinned (on a circular basis) droplets for a flat substrate undergoing tilting. This shape can be described by the Young-Laplace equation. Three formal integral equilibrium relations are derived. The interesting issue is the insertion of the polar contact angle distribution in these. The validity of the relations is checked through small Bo perturbation analysis and by comparing with numerical solution (via Surface Evolver). The significance of the integral relations in derivation of approximate solutions and on checking numerical solution is discussed.

Overall the work is sophisticated, elegant and useful. However there is a need for improvement.

-It is surprisingly that even today the simple YL equation is a research subject. Recently professor Runckestein published two papers in Langmuir discussing how the Bond number must be defined (seemingly a trivial issue). I recommend this papers to the authors.

Added to the list of References:

[GR] Berim, G. O. and Ruckenstein, E. **Bond Number Revisited: Axisymmetric Macroscopic Pendant Drop.** Langmuir, **36**, 6512-6520, 2020.

with this comment at the definition of Bo :

Consistently with [GR], our use of Bo contains only the input parameters of the drop-solid system, namely volume of the drop, gravitational acceleration, fluid density, surface tension, and radius of the drop's footprint and no data involving the dimensions of the drop unknown prior to the experiment (such as the height of the drop or radius of curvature at the drop apex).

- The circular shape of the contact line must appear in the title (the term "pinned" alone is too general).

Title: "Identities for Pinned Droplets on Tilted Surfaces" → "Identities for Droplets with Circular Footprint on Tilted Surfaces"

- I understand that "pendent" may be grammatically better but I remind you that the literature use the word "pendant" for the particular problem.

pendent → pendant

-The use of the expression "input and output parameters" although is conceivable to me, I think it will be confused for the general readers. After

all, the distinction between the parameters depends on the problem at hand. It may be replaced by "relating the problem parameters" or something like that.

The less confusing "the relevant parameters" is used

- Line 51 of page 2. This issue has also been study (semi-analytically) in:

Kostoglou, M., Karapantsios, T. Contact angle profiles for droplets on omniphilic surfaces in the presence of tangential forces. *Colloids Interfaces* **3**, 60, 2019.

for slender unpinned droplets with evolving contact line shape based on experimentally observed contact lines in

Ríos-López, I., Evgenidis, S., Kostoglou, M., Zabulis, X., Karapantsios, T.D. "Effect of initial droplet shape on the tangential force required for spreading and sliding along a solid surface" *Colloids and Surfaces A: Physicochemical and Engineering Aspects* **549**, 164-173, 2018.

The above must be discussed by the authors.

added to the list of References with the following comments:

[KK] Kostoglou, M. and Karapantsios, T. **Contact Angle Profiles for Droplets on Omniphilic Surfaces in the Presence of Tangential Forces. *Colloids Interfaces*, **3**, 60, 2019.**

[REKZK] Ríos-López, I., Evgenidis, S., Kostoglou, M., Zabulis, X., Karapantsios, T.D. **Effect of initial droplet shape on the tangential force required for spreading and sliding along a solid surface. *Colloids and Surfaces A: Physicochemical and Engineering Aspects* **549**, 164-173, 2018.**

Comments (after line 51 of page 2):

This issue has also been study (semi-analytically) in [KK], where the problem of understanding the contact line evolution of slender unpinned droplets under arbitrary scenarios of forces is addressed, based on experimentally observed contact lines.

Related to this point, in [REKZK], sessile droplets at different tilting angles are experimentally subject to varying centrifugal forces in order to explore their spreading/sliding behavior for different volumes and initial shapes (including non-axisymmetric). In particular, a test of the applicability of the Furmidge equation for the retention force is discussed.

-I do not think that the description of Surface Evolver as finite element method (which has a very specific meaning in numerical analysis) is appropriate.

The Surface Evolver software actually works as a vertex-edge-facet element model (the text has been modified accordingly).

-Use "dimensionless" instead of "a dimensional. In general there is a need for improvement in the syntax and language.

adimensional → **dimensionless.** everywhere

-I noticed that there is no a verification ratio for the identity 3 as for the other two. Also there is no validation for this identity in comparing with numerical solution.

A similar verification ratio is defined for the torque balance identity at the end of Sec.3 (Eq 40). In the limited time for revision we could provide and add numerical tests for of the torque balance identity at the end of Section 5 the title of which was changed into “Numerical check of identities”

-I would prefer to see actual contact angles (no cosines) in figs 6 and 7. Is the bimodality appeared physically correct or it is a numerical error?

Of course the contact angles are more illustrative. But the cosines of the contact angles are the numbers appearing in the equations and so more appropriate to the present work. The bimodality is not a numerical error.

- The lines 45-55 are quite inappropriate for the conclusions section and must be removed.

The corresponding lines have been removed.

-Finally the problem analyzed should not be seen as an abstractive theoretical construction. It is a real problem having an experimental counterpart (e.g. putting the droplet on a disc shaped asperity (step) on the substrate). This must be noticed.

Added in the conclusion:

We stress that the problem analyzed here should not only be seen as an abstract theoretical construction. It is a real problem having an experimental counterpart (e.g. putting the droplet on a disc shaped asperity (step) on the substrate).

• *Reviewer: 2*

Comments to the Author(s)

The authors present three identities pertaining a drop resting on an inclined surface. The identities stem from the fact that the net force and torque acting on a drop at rest must be zero. The authors not only provide a mathematical derivation of such identities, they also provide numerical and analytic (perturbative) checks. I find the paper novel and very interesting. I also find the paper well written and easy to follow.

I have just a few questions that I would like the authors to answer.

(1) My main question relates to the roles played by hydrostatic pressure and the force N of Eq. (15). Hydrostatic pressure is usually understood as coming from the weight of the liquid above a point. Hence, when the authors calculate the force N in Eq. (15), shouldn't that force be given by the weight of the drop, since $p(x)$ should just be due to the weight of the liquid above point x ?

From Eq. (17), besides the weight, N has a contribution stemming from the capillary force (as from (14)).

(2) Related to (1): In analogy to the solid-solid case, one may think that there actually should exist a force N , but that force should be the normal force (i.e., the reaction force of the surface on the drop). That normal force N should be determined from Eq. (17), but it should be not known a priori. Why is not there a normal force acting on the drop that is analogous to the solid-solid normal force?

In fact it is a good analogy, with the parallel capillary force playing the role of the static friction force.

(3) Related to (1) and (2): For a pendant drop that is hanging from the surface, there should be no hydrostatic pressure acting on the surface, and hence the weight is equal to the capillary force, that is, N should be zero in the pendant case. Is that so?

The equations in the pendant case are the same as for the sessile, with different ranges of the variables (in particular α). So N is not 0 in general. Simply compare the action on the surface at the same height but outside the droplet.

(4) Shouldn't the effect of the Laplace pressure be included in a similar way to the hydrostatic pressure? If the answer is yes, would that lead to another normal force that should be added to the N of Eq. (16)? If the answer is no, why not?

We use Eq. (20) at point 'o' but a similar equation holds at any point of the interface.

The Laplace pressure (as a difference of pressures) is a balance of forces already included (pressure and capillarity)

(5) Their numerical simulations are very nice. However, I have a minor concern. The triple line of many common liquids (e.g., water) at the angle of inclinations used in the simulations (45, 90, and 135) would not remain circumferential. Do the authors know if their results change drastically if the triple line is not circumferential? (I know the authors mention in the conclusions that this is an open problem, I am just curious to know if they have any numerical results on this.)

Actually we don't have any data on that. We invite you to read some very recent work in this direction in (see also the References therein, especially the ones authored by ElSherbini et al):

Ravazzoli, Pablo D., Cuellar, Ingrith, Gonzalez, Alejandro G. and Diez, Javier A. Contact-angle-hysteresis effects on a drop sitting on an incline plane. Phys Rev E. 99. 043105, 2019.

doi = {10.1103/PhysRevE.99.043105}

(6) Finally, a minor thing. The authors use the coefficients $C1$, $A0$ and $A1$ that are part of two Fourier expansions. However, it seems to me that their results could be obtained without ever making any reference to such Fourier expansions. Is that so? If not, do the other coefficients of such Fourier expansions play any role in these (or any other) drop identities?

The integral on the circular boundary puts the Fourier expansion into the game. Another boundary would require another expansion.

About other coefficients: you are right. In fact infinite number of data-points for the contact-angle on the boundary are encoded in the infinite number of Fourier coefficients. On the other hand, one may derive an infinite number of Force and Torque balance conditions for an infinite number of pieces of the droplet, specially those taking part in the boundary; like spherical wedges (slices) with various sizes.

Then the balance condition for the wedges are involved by other combination of the Fourier coefficients. Of course, the set of all infinite number of identities is equivalent to satisfying YL equation on every droplet's surface.

We would like to warmly thank our Referees and the Editor(s)-in-Charge for careful reading of the manuscript and for their significant suggestions for further improvements. We hope the amended version will now fit the overall recommendations.

The Authors.